# Stronger Correlations between Neurophysiological and Peripheral Disease Biomarkers Predict Better Prognosis in Two Severe Diseases

**DOI:** 10.3390/jcm9010026

**Published:** 2019-12-20

**Authors:** Yori Gidron, Marijke De Couck, Tatjana Reynders, Raphael Marechal, Sebastiaan Engelborghs, Marie D’hooghe

**Affiliations:** 1Department of Nursing, University of Haifa, Haifa 3498838, Israel; 2Center for Neuroscience, The Free University of Brussels—VUB, 1090 Brussels, Belgium; marijke.de.couck@vub.ac.be (M.D.C.); tatjana_reynders@live.be (T.R.); Sebastiaan.Engelborghs@uzbrussel.be (S.E.); marie.dhooghe@mscenter.be (M.D.); 3Department of Neurology, University Hospital of Antwerp, 2650 Edegem, Belgium; 4Department of Gasterontology, The Free University of Brussels—ULB, 1070 Brussels, Belgium; raphael.marechal@erasme.ulb.ac.be

**Keywords:** neurophysiology, cancer, prognosis, biomarkers, multiple sclerosis, brain-body synchronization

## Abstract

‘Mind–body’ debates assume that better brain–body associations are healthy. This study examined whether degree of associations between a neurophysiological vagal nerve index and peripheral disease biomarkers predict prognosis in pancreatic cancer (PC) and multiple sclerosis (MS). Sample 1 included 272 patients with advanced PC. Sample 2 included 118 patients with MS. We measured the vagal nerve index heart rate variability (HRV) derived from electrocardiograms. We examined associations between HRV and patients’ peripheral disease biomarkers: CA19-9 in PC and neurofilament light chain (NFL) in MS. Associations between HRV and each biomarker were examined separately in patients who survived or died (PC), and in those with and without relapse during 12 months (MS). In PC, HRV was significantly inversely related to the tumor marker CA19-9 in patients who later survived (*r* = −0.44, *p* < 0.05) but not in those who died (*r* = 0.10, NS). In MS, HRV was significantly and inversely related to NFL only in those who did not relapse (*r* = −0.25, *p* < 0.05), but not in those who relapsed (*r* = −0.05, NS). The degree of association between a neurophysiological vagal marker and peripheral disease biomarkers has prognostic value in two distinct diseases.

## 1. Introduction

For centuries, scholars have debated “mind–body” issues. Such debates include questions such as: does the brain (“mind”) control body processes? Does the brain know whether we are ill? One common implicit assumption stemming from such a debate is that the degree of brain–body association and synchronization may influence health and illness. Cortical regulation of body processes is important for behavioral and emotional adaptation [1]. Furthermore, Cohen and colleagues show how a weak correlation between hormonal levels and leukocytes (reflecting corticosteroid resistance in immune cells) has implications for developing colds [2]. Without such regulation of inflammation, autoimmune diseases, asthma, and cardiovascular diseases may occur [3].

The scientific field of neuroimmunology has revealed bi-directional brain-immune pathways, including vagal nerve communication of peripheral inflammatory signals and regulation of peripheral inflammation, called the vagal anti-inflammatory reflex [4]. Vagal nerve activity is indexed by heart rate variability (HRV), since both are profoundly as well as causally related (*r* = 0.88) [5]. Although mental states, genes, and lifestyle (e.g., smoking, diet, and physical activity) influence HRV [6], HRV primarily reflects cardiac vagal tone [7]. HRV was found to strongly and independently predict prognosis after myocardial infarctions, in a meta-analysis of 21 studies [8]. Similarly, HRV is an independent predictor of cancer prognosis [9]. Furthermore, since stronger vagal nerve activity inhibits three biological disease-contributors (inflammation, oxidative stress, and sympathetic hyper-activity), since vagal activity (HRV) predicts disease onset and progression, and since HRV is also related to better control of life-style risk factors (smoking, diet, exercise), it has been recently proposed that the vagal nerve protects against the major global burden of diseases [10]. One important meta-analysis revealed crucial brain regions associated with HRV, which can control behavior and peripheral physiology [7]. Of relevance, one study found that baseline HRV determined the degree of associations between brain activity and peripheral immunity (natural killer cells) and hormones (ACTH) [11], strongly showing the importance of vagal activity for brain–body synchronization. Thus, the vagus may be crucial for brain–periphery associations and synchronization. However, does a degree of neurophysiology–periphery association have prognostic value in major diseases? In the domain of chronobiology, misalignment of central and peripheral rhythms is considered to be unhealthy [12].

One way to index neurophysiology–periphery associations is by measuring the strength of associations between regulatory neurophysiological functions such as vagal nerve activity, indexed by HRV, and peripheral disease-biomarkers. If such a regulatory neurophysiological index as HRV is negatively associated with a peripheral disease marker, a stronger association may reflect better central regulation of the illness, and thus predict better prognosis. This study preliminarily examined the prognostic value of the strength of associations between neurophysiological and peripheral disease biomarkers with health outcomes. HRV reflected the neurophysiological (vagal) regulatory marker. CA19-9 and neurofilament light chain (NFL) reflected peripheral disease biomarkers in pancreatic cancer (PC) and multiple sclerosis (MS), respectively. They reflect routinely obtained biomarkers of each disease. We hypothesized that the strength of association between HRV and a peripheral index of disease burden would be greater in patients who later had good, versus poor, prognosis. We examined this question by reanalyzing data from two studies reflecting diverse diseases: PC and MS.

## 2. Methods

Participants: Sample 1 included *n* = 272 patients diagnosed with PC, from Brussels, Belgium. Among them, 52.8% had locally advanced cancer, and 47.2% had metastatic PC. Their mean (SD) age was 60.0 (11.5) years and 48.8% were women. Sample 2 included *n* = 118 patients with MS, from Belgian MS centers. Their mean (SD) age was 46.7 (9.2) years and 64% were women. Obtaining patients’ data from medical charts was approved by the medical centers’ ethical committees for each sample separately. 

Measures: In both samples, we obtained relevant background information, including patients’ age, gender, treatments, and disease phenotype (sub-groups in MS). In sample 1, we also included the inflammatory marker C-reactive protein (CRP). 

We then examined neurophysiological–periphery associations, as follows. For the neurophysiological index, we retroactively obtained brief electrocardiograms (ECGs) of the PC patients (10 s duration) and of the MS patients (5 min duration) for each patient, from which we derived patients’ HRV. Strong associations between 10 s and 5 min ECG-based HRV parameters have been demonstrated [13]. In the PC ECGs, unreadable ECGs were excluded. In the MS EGCs, middle 5 min segments of 10 min recordings were selected, omitting ectopic beats by a dedicated software. 

We specifically derived the standard deviation of intervals between normal heartbeats (SDNN), reflecting mostly vagal activity. As biomarkers of peripheral disease, levels of patients’ tumor marker CA-19-9 were examined in PC. Plasma neurofilament light chain (NFL) levels were determined by single molecule array immunoassay as the peripheral marker of neuronal damage in MS. Concerning prognosis, we examined overall survival in PC and the 12 month relapse risk in MS. 

Statistical analyses: To statistically examine whether the degree of association between the neurophysiological–peripheral disease indexes had prognostic value, we retroactively examined in each disease the associations between HRV and the peripheral disease biomarker (CA-19-9 in PC, NFL in MS) in each prognostic group separately. Thus, we examined these correlations in PC in patients who survived versus not survived, and in MS, in patients who relapsed versus non-relapsed later, separately. These were partial correlations, adjusting for important confounders mentioned below.

## 3. Result

In the full PC sample, there were 29 alive patients and 197 dead at follow-up. To make the model parsimonious, we enabled significant confounding predictors of death to statistically “compete” in a logistic regression, which yielded surgery, chemotherapy, and age as confounders, whose effects were subsequently adjusted for. In the full sample, the correlation between HRV and CA-19-9 was *r* = −0.01 (NS), adjusting for age, surgery, and chemotherapy. However, in those who later survived, this partial correlation was negative and significant: *r* = −0.44 (*p* < 0.05). In contrast, this partial correlation was not significant in those who later died (*r* = 0.10, NS; see Figure 1a,b). Interestingly, adding CRP as a covariate to the first correlation made it no longer statistically significant (*p* = 0.06) suggesting that reduced inflammation partly mediated the correlation between HRV and CA19-9 in alive patients. 

In MS patients, 18 patients relapsed and 95 did not at the 12 month follow-up. Though no confounder predicted relapse, the MS sub-group was subsequently entered as a control variable, to more rigorously adjust for its effects. HRV was inversely related to NFL in the full sample (*r* = −0.19, *p* < 0.05). In patients who did not relapse, this partial correlation was significant: *r* = −0.25 (*p* < 0.05), whereas this partial correlation was lost and not significant in patients who later relapsed (*r* = −0.05, NS), in both sub-samples, independent of MS sub-group (see Figure 2a,b). This pattern was precisely the same when using the fully vagal dependent HRV time domain parameter of RMSSD. In those without relapse, HRV was inversely related to NFL (*r* = −0.25, *p* < 0.05), but not in those with relapse (*r* = −0.11, NS).

## 4. Discussion

This may be the first study showing that the degree of association between a neurophysiological index (HRV) and a peripheral disease biomarker appears to be related to prognosis in two distinct diseases. Specifically, we found negative significant correlations between the vagal nerve index HRV and disease burden (CA19-9 in PC, NFL in MS), only in patients with a better outcome, while no such correlations were seen in patients with poor outcomes (later died in PC or relapsed in MS). These results echo the known regulatory roles of the vagus [4] and its potential protective roles in major diseases [10]. These results also support numerous studies showing that higher HRV independently predicts better prognosis after myocardial infarctions [8] and in cancer [9]. Finally, these results echo those showing that lacking hormonal-immune (cortisol-leukocyte) associations predicts developing colds [2]. However, our findings observed in the present study extend past studies to suggest that stronger negative neurophysiology–peripheral disease associations may reflect better neurophysiological regulation of disease burden, which subsequently predicts better prognosis in two distinct illnesses, PC and MS. The exploratory analysis, including CRP, in PC patients suggests that vagal modulation of inflammation may partly account for these results, yet this requires further verification. These results also show the importance of neuromodulation in diseases, with a focus on the vagal nerve’s regulatory roles. 

Limitations of this study include not having direct brain activity measures, using self-reports for relapse in MS, having few relapsing cases in MS and few alive patients in PC, and not adjusting for patients’ mental state. Furthermore, the ECG durations (especially in PC patients) were short and this did not enable to measure the frequency domain HRV index of high-frequency HRV, especially in PC. However, our observation in MS patients that both SDNN and RMSSD showed identical patterns points at possible specific vagal regulation of peripheral disease biomarkers. Additionally, the observed neurophysiology–periphery correlations may not reflect temporal or functional brain–periphery synchronization. Future studies should address these limitations by measuring neurophysiological and peripheral disease indexes simultaneously, or even better, in an experimental setting, and then examine prognosis. Nevertheless, the similar pattern of results found in the present study in two distinct and severe diseases suggests a possible general phenomenon. However, the generality of these observations needs to be examined in other conditions as well. 

Various non-invasive forms of neuromodulation, such as non-invasive vagal nerve stimulation, have regulatory effects such as reducing peripheral inflammation [14]. If the results presented here will be replicated, effects of interventions aimed at increasing neurophysiology–periphery regulation and associations (using neurofeedback, HRV biofeedback, vagal nerve stimulation) on health outcomes could be tested in randomized controlled trials. 

## Figures and Tables

**Figure 1 jcm-09-00026-f001:**
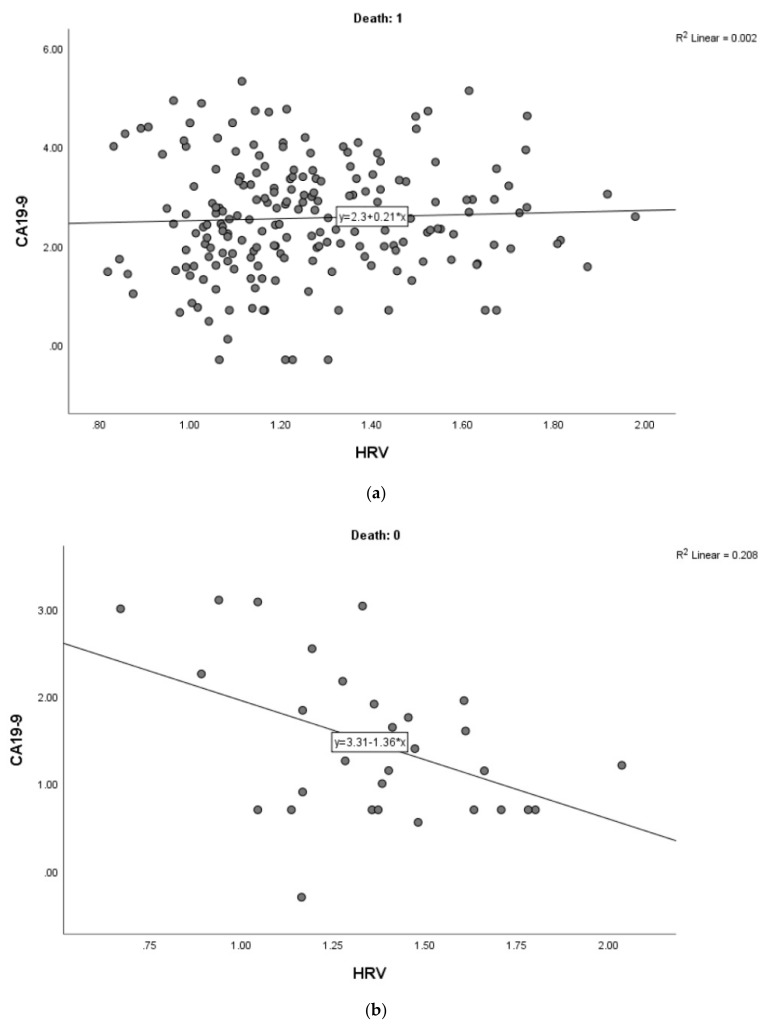
(**a**) Relationship between heart rate variability (HRV) and the tumor marker CA19-9 in patients with pancreatic cancer who later died (correlation without adjustment for confounders mentioned in text); (**b**) Relationship between heart rate variability (HRV) and the tumor marker CA19-9 in patients with pancreatic cancer who survived (correlation without adjustment for confounders mentioned in text).

**Figure 2 jcm-09-00026-f002:**
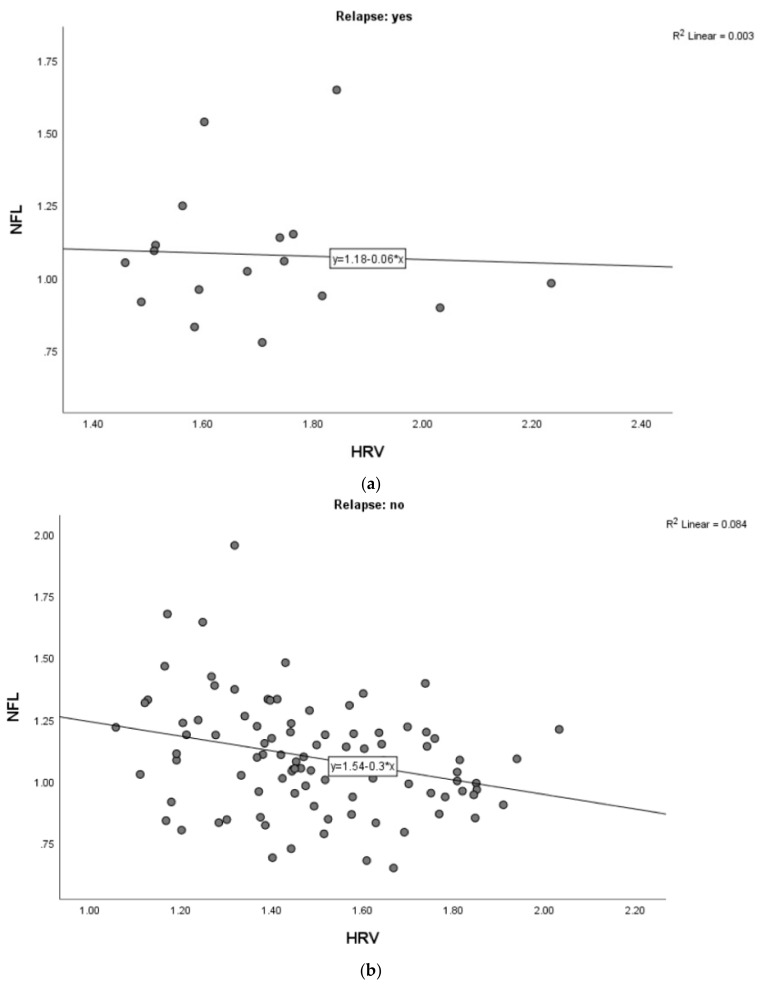
(**a**) Relationship between heart rate variability (HRV) and the multiple sclerosis (MS) marker NFL in patients who did relapse (correlation without adjustment for confounder mentioned in text); (**b**) Relationship between heart rate variability (HRV) and the MS marker neurofilament light chain (NFL) in patients who did not relapse (correlation without adjustment for confounder mentioned in text).

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
