# Peer review of "Stronger Correlations between Neurophysiological and Peripheral Disease Biomarkers Predict Better Prognosis in Two Severe Diseases"

_jcm, 2019, doi:10.3390/jcm9010026_

Round 1

Reviewer 1 Report

Dear Editor,

            Thanks for the possibility to review the manuscript titled “Stronger correlations between neurophysiological and peripheral disease biomarkers predict better 3 prognosis in two severe diseases”. I think that the manuscript addresses an issue that is very important and relevant for the researchers that study neurophysiology-periphery associations and their possibility to predict prognosis.

Summary

The authors start introducing the “mind-body” debate, then, the authors focus mainly on the neurophysiology-periphery associations and on their possibility to predict the prognosis in two diseases: in pancreatic cancer (PC) and multiple sclerosis (MS).

In the introduction section they also present studies in which the vagal nerve activity was indexed by heart rate variability (HRV) and used to predict prognosis in other diseases.

Afterwards, the authors present the underlying logic on which the work is based: bi-directional brain-immune pathways that include vagal nerve communication of peripheral inflammatory signals and regulation of peripheral inflammation, called the vagal anti-inflammatory reflex. Since in the domain of chronobiology, misalignment of central and peripheral rhythms is considered to be unhealthy, the authors ‘ aim is to consider the prognostic value of the strength of associations between neurophysiological-peripheral disease biomarkers and health outcomes in both PC and MS.

Main impressions

I think that the article addresses an important issue and it is interesting. It can be useful to help the researchers that study the prognostic value of the strength of associations between neurophysiological-peripheral disease biomarkers and health outcomes.

Introduction

The introduction has a clear logical progression. The text is well organized.

Method

The sampling is appropriate. I appreciated the sample size.

The author accurately explain how the data were collected

The materials been adequately described

The authors clearly describe procedure. The statistic is appropriate.

Results and discussion

The authors clearly describe the results by presenting the partial correlation between neurophysiological and peripheral biomarkers in subjects who later survive and in subjects who later died in PC, and in patients that did not relapse and who later relapsed in MS.

Since Figures 1a-b depict the correlations between HRV and CA19-91 9 for PC patients, I think that the authors has to add Figure 2a-b to show graphically the same correlation for MS patients.

I appreciated finally the large section dedicated to the limitations of the study.

Originality/Novelty:  well defined Significance:  the results are interpreted appropriately Quality of Presentation: the article is written in an appropriate way? The data and analyses presented are appropriate Scientific Soundness: the study is correctly designed and technically sound Interest to the Readers: the conclusions interesting for the readership of the Journal English Level: The English language is appropriate and understandable, but I am not an English native speaker.

Author Response

A figure of the relationship between HRV and NFL was added - please see Figures 2a-b.

Reviewer 2 Report

I found this brief report to be quite interesting, as it addresses a novel aspect of mind-body research, namely variability within the mind-body connection (fascinating!) and potential health consequences.

I am mindful of the space constraints of brief reports, but think some changes can be made to improve its impact.

1) Lines 32 and 33: The authors state that a "common implicit assumption" of mind-body research is that the degree of synchronization can impact health. Can the authors provide citation(s) supporting this? 

2) Line 37: what other factors influence HRV, beyond vagal activity? E.g., emotional state, physical activity? In general, this manuscript treats HRV and vagal activity as strongly related, but a consideration of other sources of HRV is necessary, including in discussing study limitations (i.e., no other sources of HRV is considered; could it be that HRV is tracking emotional state and that is what is linked with improved health/survival rather than vagal activity?)

3) Line 40: Please clarify: "the vagus nerve inhibits three biological..." Presumably this means vagal nerve activity?

4) Line 43: Citation #5 is relied on heavily (implicitly and explicitly) to support the idea that variation in vagal activity underlies important health outcomes. Are there critiques of this citation or idea that the authors can cite to provide balance?

5) Are there any meaningful differences between a 10 second and 5 minute ECG when deriving HRV that should be discussed?

6) Methods: Please provide n's for survived/deceased and relapsed/non-relapsed patients, as these numbers are mentioned as a limitation to the study in the Discussion section.

7) Results: The authors state that their results are partial correlations, adjusted for important confounders. However, the only confounders reported are surgery and chemotherapy in pancreatic cancer patients. What confounders were there in MS patients? Is it possible to adjust for other sources of HRV in this dataset? If not, this should be addressed in the limitations.

8) Results: If space permits, please also present figures showing correlations in relapsing/non-relapsing MS patients.

Author Response

1. Lines 32 and 33: The authors state that a "common implicit assumption" of mind-body research is that the degree of synchronization can impact health. Can the authors provide citation(s) supporting this?

Reply: We thank the reviewer for this important comment. We added this paragraph to answer it:

Cortical regulation of body processes is important for behavioral and emotional adaptation (10). Furthermore, Cohen and colleagues show how a weak correlation between hormonal levels and leukocytes (reflecting corticosteroid resistance in immune cells) has implications for developing colds (7). Without such regulation of inflammation, autoimmune diseases, asthma and cardiovascular diseases may occur (11).

2) Line 37: what other factors influence HRV, beyond vagal activity? E.g., emotional state, physical activity? In general, this manuscript treats HRV and vagal activity as strongly related, but a consideration of other sources of HRV is necessary, including in discussing study limitations (i.e., no other sources of HRV is considered; could it be that HRV is tracking emotional state and that is what is linked with improved health/survival rather than vagal activity?)

Reply; We thank the reviewer for this question and thus added this clarification.

Vagal nerve activity is indexed by heart rate variability (HRV), since both are profoundly as well as causally related (r = 0.88; 2). Though mental states, genes and life-style (e.g., smoking, diet and physical activity) influence HRV (12-13), HRV primarily reflects cardiac vagal tone (13).

3) Line 40: Please clarify: "the vagus nerve inhibits three biological..." Presumably this means vagal nerve activity?

Reply: We added “stronger vagus nerve activity inhibits three biological disease-contributors”.

4) Line 43: Citation #5 is relied on heavily (implicitly and explicitly) to support the idea that variation in vagal activity underlies important health outcomes. Are there critiques of this citation or idea that the authors can cite to provide balance?

Reply: Citation 5 is cited only twice in the manuscript. Nevertheless, we added additional references to support it. One important meta-analysis revealed crucial brain regions associated with HRV, which can control behavior and peripheral physiology (14).

5) Are there any meaningful differences between a 10 second and 5 minute ECG when deriving HRV that should be discussed?

Reply: We added this information to the methods section: Strong associations between 10-second and 5-min ECG based HRV parameters have been demonstrated (15).

6) Methods: Please provide n's for survived/deceased and relapsed/non-relapsed patients, as these numbers are mentioned as a limitation to the study in the Discussion section.

Reply: These numbers were provided in the results section per sample.

7) Results: The authors state that their results are partial correlations, adjusted for important confounders. However, the only confounders reported are surgery and chemotherapy in pancreatic cancer patients. What confounders were there in MS patients? Is it possible to adjust for other sources of HRV in this dataset? If not, this should be addressed in the limitations.

Reply: We added this information in the Results section:

To make the model parsimonious, we enabled significant confounding predictors of death to statistically “compete” in a logistic regression, which yielded surgery, chemotherapy and age as confounders, whose effects were subsequently adjusted for.

Though no confounder predicted relapse, MS sub-group was subsequently entered as a control variable, to more rigorously adjust for its effects.

8) Results: If space permits, please also present figures showing correlations in relapsing/non-relapsing MS patients.

Reply: This figure was added (see Figures 2a-b).

Reviewer 3 Report

The intro was well-written; however, the connection between mind-body issues and vagal activity in this manuscript is weak. In the introduction, inflammatory signals is suggested as a connecting factor.  However, there are no such measures in this report.  Including a factor(s) would significantly increase the strength of the paper and, at the very least, justify the content within the abstract (lines 13-14, and 30-36). Line 47 should be revised for clarity (i.e., ‘lacking’ is probably not the right word choice here). Same could be said for line 103. The prognostic markers, CA-19-9 and NFL should be briefly discussed in the introduction. What is the rationale for using these markers? Why was only 10 sec of ECG analyzed for PC, but then there is 5-min for MS patients? Ten seconds (i.e., ~10 beats) is quite short considering that SDNN is more accurate when calculated over longer ECG segments.  What criteria was used in selecting a representative ECG segment in a patient? And, how were ectopic beats handled?  Describing the criteria that were used is important to the validity of the research findings. Considering that both SNS and PNS activity contribute to SDNN, using additional markers of vagal activity would be helpful in strengthening the conclusions made in the paper (e.g., low frequency band power, 2V using symbolic dynamics, or more in the time domain). Thus, using longer segments and set criteria for segmenting ECG would be helpful to the PC analysis. The graphs are small and blurry and thus are difficult to interpret. There should be graphs representing the NFL response.  It would also be helpful to include a variability measure such as standard error of the estimate with the regression analysis. What was the ratio of men to women in the MS group?

Author Response

The intro was well-written; however, the connection between mind-body issues and vagal activity in this manuscript is weak.

Reply: We cited a meta-analysis revealing associations between brain and vagal activity in regions important for behavioral and physiological regulation and emphasized the regulatory and bridging role of the vagus in brain-body communication.

In the introduction, inflammatory signals is suggested as a connecting factor.  However, there are no such measures in this report.  Including a factor(s) would significantly increase the strength of the paper and, at the very least, justify the content within the abstract (lines 13-14, and 30-36).

Reply: We added to the Methods section the measurement of the inflammatory marker C-reactive protein (CRP). In addition, we added to the results this new finding:

Interestingly, adding CRP as a covariate to the first correlation made it no longer statistically significant (p = 0.06) suggesting that reduced inflammation partly mediated the correlation between HRV and CA19-9 in alive patients. We also referred to this in the Discussion.

Line 47 should be revised for clarity (i.e., ‘lacking’ is probably not the right word choice here).

Reply: We omitted the term lacking in that context.

Same could be said for line 103. The prognostic markers, CA-19-9 and NFL should be briefly discussed in the introduction. What is the rationale for using these markers?

Reply: We added before the hypothesis these sentence:

HRV reflected the neurophysiological regulatory marker. CA19-9 and neurofilament light chain (NFL) reflected peripheral disease biomarkers in pancreatic cancer (PC) and multiple sclerosis (MS), respectively. They reflect routinely obtained biomarkers of each disease. 

Why was only 10 sec of ECG analyzed for PC, but then there is 5-min for MS patients? Ten seconds (i.e., ~10 beats) is quite short considering that SDNN is more accurate when calculated over longer ECG segments. 

Reply: We added in the Methods section information on the reliability of 10-sec HRV measures when compared to 5-min.

What criteria was used in selecting a representative ECG segment in a patient? And, how were ectopic beats handled?  Describing the criteria that were used is important to the validity of the research findings.

Reply: We added in the Methods section the following: In the PC ECGs, unreadable ECGs were excluded. In the MS EGCs, middle 5-min segments of 10-min recordings were selected, omitting ectopic beats by a dedicated software. These issues were added to the Discussion as limitations and future directions as well.

Considering that both SNS and PNS activity contribute to SDNN, using additional markers of vagal activity would be helpful in strengthening the conclusions made in the paper (e.g., low frequency band power, 2V using symbolic dynamics, or more in the time domain). Thus, using longer segments and set criteria for segmenting ECG would be helpful to the PC analysis.

Reply: We added to the Results section on MS the following: This pattern was precisely the same when using the fully vagal dependent HRV time domain parameter of RMSSD: In those without relapse, HRV was inversely related to NFL (r = -0.25, p < 0.05) but not in those with relapse (r = -0.11, NS).

The graphs are small and blurry and thus are difficult to interpret. There should be graphs representing the NFL response. 

Reply: We increased the size of the figures and added a figure for the HRV-NFL relations as well. See Figures 2a-b.

It would also be helpful to include a variability measure such as standard error of the estimate with the regression analysis. What was the ratio of men to women in the MS group? 

Reply: The main statistical analyses were partial correlations, hence standard errors are not relevant. The percentages of women were added in the Methods section for both samples. 

Round 2

Reviewer 3 Report

The authors substantive revisions to the manuscript are appreciated.  The clarity of the figures still needs to be addressed.  The axes of the figures are pixelated and unreadable.  In addition, there are the following minor concerns,

Line 5-6: Inconsistency in commas and PhD in the names

Line 11:  'corresponding' rather than "correponding"

Line 45:  Is 'Although mental states...' intended here rather than "Though"?

Line 48:  Run-on sentence and sentence structure of the sentence beginning with, "Furthermore, because".

Line 57-58:  Consistency in spelling 'synchronization'.

Figure captions:  it would be helpful to include the variables that you controlled for in the caption.

Author Response

The authors substantive revisions to the manuscript are appreciated.  The clarity of the figures still needs to be addressed.  The axes of the figures are pixelated and unreadable. 

REPLY:

The figures are now presented larger and each separated and are hence clear.

In addition, there are the following minor concerns,

Line 5-6: Inconsistency in commas and PhD in the names

REPLY: Corrected and degrees were omitted.

Line 11:  'corresponding' rather than "correponding"

REPLY: Thank you. Corrected!

Line 45:  Is 'Although mental states...' intended here rather than "Though"?

REPLY: Changed as suggested.

Line 48:  Run-on sentence and sentence structure of the sentence beginning with, "Furthermore, because".

REPLY: This sentence was made clearer, thank you!

Line 57-58:  Consistency in spelling 'synchronization'.

REPLY: All are now uniformly written with a "z".

Figure captions:  it would be helpful to include the variables that you controlled for in the caption.

REPLY: The SPSS graphic section does not enable to perform partial correlations, rather only simple bivariate correlations. This was indicated in each figure.